# Supramolecular Enzymatic Labeling for Aptamer Switch-Based Electrochemical Biosensor

**DOI:** 10.3390/bios12070514

**Published:** 2022-07-12

**Authors:** Anabel Villalonga, Concepción Parrado, Raúl Díaz, Alfredo Sánchez, Beatriz Mayol, Paloma Martínez-Ruíz, Diana Vilela, Reynaldo Villalonga

**Affiliations:** Nanosensors and Nanomachines Group, Department of Analytical Chemistry, Faculty of Chemistry, Complutense University of Madrid, 280409 Madrid, Spain; anabelvi@ucm.es (A.V.); cparrado@ucm.es (C.P.); rauldi01@ucm.es (R.D.); alfredos@ucm.es (A.S.); beamayol@ucm.es (B.M.); palmarti@quim.ucm.es (P.M.-R.)

**Keywords:** aptamer, biosensor, carcinoembryonic antigen, cyclodextrin, gold nanostars

## Abstract

Here we report a novel labeling strategy for electrochemical aptasensors based on enzymatic marking via supramolecular host–guest interactions. This approach relies on the use of an adamantane-modified target-responsive hairpin DNA aptamer as an affinity bioreceptor, and a neoglycoconjugate of β-cyclodextin (CD) covalently attached to a redox enzyme as a labeling element. As a proof of concept, an amperometric aptasensor for a carcinoembryonic antigen was assembled on screen-printed carbon electrodes modified with electrodeposited fern-like gold nanoparticles/graphene oxide and, by using a horseradish peroxidase-CD neoglycoenzyme as a biocatalytic redox label. This aptasensor was able to detect the biomarker in the concentration range from 10 pg/mL to 1 ng/mL with a high selectivity and a low detection limit of 3.1 pg/mL in human serum samples.

## 1. Introduction

The design of novel affinity-based electrochemical biosensors with improved sensitivity is highly related to the employment of efficient methods for signal production and amplification [1]. Generally, such methods rely on the use of secondary or complementary biological receptors labeled with enzymes, nanomaterials, and redox probes [2,3,4]. For the special case of aptamers, which can fold in tridimensional structures able to refold upon binding to the target molecule, labeling can be performed at the nucleic acid end group not linked to the electrode. Such an approach allows establishing one-pot sensing strategies, but also the assembly of more complex architectures for signal enhancement without using a secondary bioreceptor [5].

Labeling for enhanced signal production in electrochemical biosensors and other analytical bioassays can be performed through covalent or non-covalent linkages. Among them, the biotin-(strept)avidin conjugation system is by far the most commonly employed [6,7]. This fact is based on the intrinsic advantages of this labeling system, such as the relatively small size of the biotin and streptavidin/avidin molecules, allowing extensive binding to bioreceptors without affecting their affinity properties due to steric hindrance. In addition, avidin and streptavidin rapidly bind biotin with a high specificity and affinity, with association rates and affinity constants around 10^7^ M^−^^1^ s^−^^1^ and 10^13^ M^−^^1^ s^−^^1^, respectively [8]. Such binding power allows the use of this labeling system in extreme bioassay conditions of pH, temperature, ionic strength, and the presence of organic solvents or denaturing agents [9]. Finally, avidin and streptavidin can bind up to four biotin molecules allowing the assembly of advanced 3D architectures for signal enhancement.

However, biotin interference can produce serious analytical distortions in clinical assays using the biotin-(strept)avidin labeling approach [10]. During the last years, this problem has increased due to the establishment of new biotin-based therapeutic protocols for multiple diseases, as well as the inclusion of this vitamin in several nutritional products [11]. That is why the development of alternative labeling approaches for electrochemical affinity biosensors and other clinical assays receives considerable attention. Here we describe a supramolecular strategy for the enzymatic labeling of aptamers based on the ability of β-cyclodextrin (CD) moieties, covalently linked to the redox enzyme peroxidase, to form stable inclusion complexes with hydrophobic molecules. As a proof-of-concept, this approach was employed to construct an amperometric aptasensor for carcinoembryonic antigen (CEA) by using an adamantane-modified DNA aptamer (Apt-ADA) as biorecognition element and screen-printed carbon electrodes (SPE) coated with electrodeposited graphene oxide (GO) and fern-like gold nanoparticles (AuNF) as a transduction interface. CDs have been largely employed as a scaffold to assemble catalytic and affinity biosensors [12,13], but to our knowledge, their use as substitutes of the biotin-(strept)avidin labeling system for electrochemical aptasensors has not been described.

## 2. Materials and Methods

### 2.1. Reagents and Instruments

SPE and GO were purchased from Orion High Technologies. The specific anti-CEA aptamer modified with thiol and primary amino groups at the 5′ and 3′ends, respectively, (Apt, 5′-HS-(CH_2_)_6_-CCACGATACCAGCTTATTCAATTCGTGG-(CH_2_)_6_-NH_2_-3′) [14], N-hydroxysuccinimide (NHS), horseradish peroxidase (HRP), tris(2-carboxyethyl)phosphine hydrochloride (TCEP), 1-ethyl-3-(3-dimethylaminopropyl)carbodiimide hydrochloride (EDAC), phosphate buffered saline with 3% (*w*/*v*) non-fat milk, pH 7.4 (blocking buffer), 2,2′-azino-bis(3-ethylbenzothiazoline-6-sulphonic acid) (ABTS), tetrachloroauric (III) acid (HauCl_4_), and the other reagents were acquired from Sigma-Aldrich (Madrid, Spain) Electrochemical measurements were performed with a PalmSens4 potentiostat (PalmSens, Houten, The Netherlands). High-resolution field emission scanning electron microscopy (FE-SEM) was performed with a JEOL JSM 7600F microscope (JEOL, Akishima, Japan. Spectrophotometric measurements were performed with an Ultrospec™ 8000 Dual Beam UV/VIS spectrophotometer (Biochrom Ltd., Cambridge, UK).

### 2.2. Synthesis of Chemically Modified Biomolecules

Mono-6-ethylenediamino-6-deoxy-β-cyclodextrin was synthesized as described [15]. To prepare HRP-CD, 10 mg HRP and 50 mg mono-6-ethylenediamino-6-deoxy-β-cyclodextrin were dissolved in 1.0 mL of 50 mM sodium phosphate buffer, pH 6.0, and then 15 mg EDAC and 15 mg NHS were added. The mixture was stirred for 1 h at room temperature and further on for 16 h at 4 °C. The solution was then dialyzed vs. 50 mM sodium phosphate buffer, pH 6.0 and concentrated by 10K Amicon Ultra-05 centrifugal filter (EMD Millipore Co., Madrid, Spain) to a final concentration of 10 mg protein/mL. The specific activity of HRP toward H_2_O_2_ using ABTS as reducing substrate [16], increased from 6308 U/mg to 9379 U/mg after modification with CD.

To prepare the adamantane-modified aptamer (Apt-ADA), 1.0 mg EDAC, 1.0 mg NHS and 25 µL of 24 mM 1-adamantane carboxylic acid in 0.1 M NaOH were added to 250 µL of 50 mM sodium phosphate buffer, pH 6.0. The resulting solution was then mixed with 250 µL of the same buffer containing 51 nmol Apt. The mixture was stirred for 1 h at room temperature and further on for 16 h at 4 °C. The solution was then dialyzed vs. 50 mM sodium phosphate buffer, pH 6.0, and concentrated by 3K Amicon Ultra-05 centrifugal filter to a final concentration of 100 µM Apt-ADA. Before using, the Apt-ADA molecule was folded to the hairpin structure as described [14,17].

### 2.3. Preparation of Aptamer-Functionalized Electrodes (Apt-ADA/AuNF/GO/SPE)

SPE were first coated with electrodeposited GO by dropping 50 µL of 1.0 mg/mL GO dispersion in 0.1 M sodium carbonate buffer, pH 9.0 on the electrode surface and then, applying 5 scans of cyclic voltammetry between −1.5 V and 0.6 V at 25 mV/s [17]. The electrode was then washed with MilliQ water, dried under N_2_, and coated with 60 µL of a freshly prepared aqueous solution of 0.1 M HAuCl_4_ in 1.0 M NH_4_Cl. Au nanoparticles were electrodeposited through a modified protocol [18], by using amperometry at −3.0 V during 20 s. The electrode was finally washed and dried as described.

Before immobilization, the Apt-ADA molecules were folded to the hairpin structure [14,17]. Briefly, 75 µL of 67 µM Apt-ADA solution was heated at 95 °C during 5 min followed by slow cooling to room temperature. The folded aptamer solution was further mixed with 25 µL of 30 mM TCEP and incubated at room temperature during 30 min. The working electrode was then coated with 8 µL of the resulting aptamer solution and incubated at 4 °C during 1 h. The electrode was washed with cold 100 mM NaCl, 5 mM MgCl_2_ in 20 mM Tris-HCl buffer solution, pH 7.4, dried under N_2_, and then 8 µL of 2% (*w*/*v*) casein solution in 0.1 M sodium phosphate buffer, pH 7.4, were dropped on the working electrode surface. After 1 h incubation at 4 °C, the Apt-ADA/AuNF/GO/SPE electrode was washed with 0.1 M sodium phosphate buffer, pH 7.4, dried and kept at 4 °C until use.

### 2.4. Electroanalytical Procedure

CEA samples were prepared in blocking buffer ten-fold diluted with 0.1 M sodium phosphate buffer, pH 7.4. CEA samples (8.0 µL) were dropped on the Apt-ADA/AuNF/GO/SPE working surface and kept at 4 °C during 30 min. The electrode was then washed with the same cold buffer and dried, and then 8.0 µL of 100 µg/mL HRP-CD solution in 50 mM sodium phosphate buffer, pH 6.0, was added to the working electrode. After 1 h incubation at 4 °C, the electrode was washed and dried. To drive analytical measurements, 45 µL of 100 mM sodium phosphate buffer, pH 7.4, were added to the electrochemical cell and amperometric signals were recorded at −200 mV after addition of 5 µL of a freshly prepared 50 mM H_2_O_2_ solution.

The intensities of the amperometric signals correspond to the differences between the stabilized signal and the signal after the addition of H_2_O_2_ solution. The relative amperometric signals consider the non-specific signal and it has been calculated as:$$\mathrm{Relative}\;\mathrm{Amperometric}\;\mathrm{Response}\;\%=\frac{\left(i_{\mathrm{specific}\;\mathrm{signal}}-i_{\mathrm{non}-\mathrm{specific}\;\mathrm{signal}}\right)}{i_{\mathrm{specific}\;\mathrm{signal}}}\times 100$$

### 2.5. Real Sample Analysis

Commercial lyophilized human serum sample was reconstituted with 1 mL water and then 5-fold diluted in 100 mM sodium phosphate buffer, pH 7.4, containing 0.1% (*w*/*v*) casein. The aptasensor was then tested toward CEA in this sample by employing a similar protocol as described in Section 2.4, and by using the standard addition method.

## 3. Results and Discussion

Figure 1A shows the preparation of the Apt-ADA/AuNF/GO/SPE device for CEA determination.

### 3.1. Characterization of the Electrode Surface Using Scanning Electronic Microscopy (SEM)

The bare working electrode (Figure 1A) was first coated with GO through electrodeposition of the nanomaterial by cyclic voltammetry [17]. As confirmed by SEM analysis (Figure 1B), the electrode surface was homogenously coated with crumpled nanosheets of GO.

Au nanoparticles were then electrodeposited by the amperometric reduction of HauCl_4_ at a high voltage in the presence of NH_4_Cl. Under these conditions, fern-like nanoparticles were grown on the electrode surface, as revealed by SEM (Figure 1C). The metal composition of these fronded nanostructures was confirmed by energy-dispersive X-ray spectroscopy (EDS), as is shown in Figure 1D.

### 3.2. Selection of the Materials for the Sensor Construction

The rational of using these combined nanomaterials as transduction elements is based on their potential synergic effect on the electrocatalytic reduction of H_2_O_2_, the substrate for the HRP-CD labeling element in this aptasensor. This fact was demonstrated by cyclic voltammetry of the electrode at the different nanostructuration steps, recorded in 50 mM H_2_O_2_ solutions. As is illustrated in Figure 2A, on the one hand, a slight increase in the reduction currents was observed for the SPE after electrodeposition of GO, suggesting this nanomaterial conferred electrocatalytic properties to the electrode surface. On the other hand, large catalytic currents were measured with the electrode after modification with the AuNF, demonstrating the large electrocatalytic activity of this nanomaterial toward H_2_O_2_. This improved electrocatalytic behaviour can be ascribed to the well-known intrinsic peroxidase-like activity of Au nanoparticles [19], significantly enhanced by their fronded morphology. 

The AuNF electrodeposited on the working electrode surface was employed as support for the further immobilization of the thiol and adamantane-modified aptamer via chemisorption linkages. Casein was finally adsorbed on the sensing surface to act as a blocking agent, avoiding non-specific interactions with the sample components. The assembly of this functionalized electrode was optimized by determining the effect of the different experimental conditions on the amperometric response toward 20 pg/mL CEA. Each optimization experiment was repeated three times, and the results are shown in Appendix A.

Taking into account the high electrocatalytic activity of the electrode surface for H_2_O_2_, we evaluated the amperometric response of the aptasensor toward 20 pg/mL CEA in the presence and the absence of hydroquinone as the HRP-CD co-substrate. As is illustrated in Figure 2B, the assembled aptasensor showed an amperometric response without hydroquinone, and the ratio for the signals with and without H_2_O_2_ increased from 1.2 to 1.5 in comparison with the aptasensor using hydroquinone. Accordingly, further analytical determinations were performed without this co-substrate. It is well-known that the HRP-mediated electrocatalytic reduction of H_2_O_2_ occurs through the following mechanism [20]:(1)$$\mathrm{HRP}\;\left({\mathrm{Fe}}^{\mathrm{III}}\right)+{\;H}_{2}O_{2\;}\to \;\mathrm{Compound}\;I\;\left({\left[{\mathrm{Fe}}^{\mathrm{IV}}=O\right]}^{+}\right)+{\;H}_{2}O$$
(2)$$\mathrm{Compound}\;I\;\left({\left[{\mathrm{Fe}}^{\mathrm{IV}}=O\right]}^{+}\right)+\;e\;+{\;H}^{+}\to \;\mathrm{Compound}\;\mathrm{II}\;({\mathrm{Fe}}^{\mathrm{IV}}=O)$$
(3)$$\mathrm{Compound}\;\mathrm{II}\;({\mathrm{Fe}}^{\mathrm{IV}}=O)+\;e\;+{\;H\;}^{+}\to \;\mathrm{HRP}\left({\mathrm{Fe}}^{\mathrm{III}}\right)+{\;H}_{2}O\;$$

Since compound II is more stable than the free radical compound I, the increased cathodic response of the aptasensor in the absence of hydroquinone should be produced by the electroreduction of compound II at the electrode surface through a direct electron transfer mechanism [20]. In this sense, the high density of AuNF at the electrode should allow a tuning effect between the enzyme’s active center and the sensing surface.

### 3.3. Electrochemical Characterization of the Assembling Steps

The assembling steps for this aptamer-modified electrode were characterized by cyclic voltammetry and electrochemical impedance spectroscopy (EIS), by using [Fe(CN)_6_]^4−/3−^ as a redox probe. As is illustrated in Figure 3A, the cyclic voltammograms measured for the raw, GO-, and AuNF-modified electrodes showed typical quasi-reversible patterns with well-defined diffusion-limited behaviors. In addition, the redox peak heights progressively increased after electrodeposition of the nanomaterials due to their high electroconductive properties, increasing the electroactive surface area of the electrode and thus favoring the electron transfer processes at these interfaces. In this sense, it was further demonstrated that the electroactive surface area of the electrode, calculated by applying the Randles–Sevcik equation, varied from 12.2 mm^2^ to 12.6 mm^2^ and 15.4 mm^2^ (Figure 3(Aa), (Ab) and (Ac), respectively) upon sequential electrodeposition of GO and AuNF. 

On the contrary, the immobilization of aptamer molecules caused a decrease in the redox peak heights, and accordingly, the electroactive surface area of the electrode was reduced to 13.8 mm^2^ (Figure 3(Ad)) after biofunctionalization. Further coating with casein gave rise to a sharp decrease in the peak currents, a large separation between the anodic and cathodic peaks, and a reduction of the electroactive surface area up to 8.0 mm^2^ (Figure 3(Ae)). These facts can be justified by the barrier formed by the protein on the electrode surface, suggesting a high coverage with the coating agent.

Similar results were obtained by EIS, by fitting the experimental data to a conventional Randles equivalent circuit where R_S_ is the solution resistance defining the initial *Z*′ value, R_ET_ is the charge transfer resistance defining the semicircle diameter in the Nyquist plot, C_DL_ is the double layer capacitance, and Z_W_ is the diffusional resistance (Warburg element) manifested by the line in the low frequency region. As is illustrated in Figure 3B, the semicircle diameter observed at high frequencies in the Nyquist corresponding to the bare electrode was progressively reduced from 226.9 Ω to 30.8 Ω and 5.1 Ω (Figure 3(Ba), (Bb) and (Bc), respectively) after the electrodeposition of GO and AuNF, respectively. This reduced electron transfer resistance can be ascribed to the improved electroconductive properties of the electrode surface after modification with these nanomaterials. On the contrary, surface modification with the non-conductive molecules of Apt-ADA and casein yielded a noticeable increase in the electron transfer resistance, reaching values of 52.6 Ω and 214.9 Ω, respectively (Figure 3(Bd) and (Be), respectively).

The aptamer-modified electrode was evaluated for the amperometric detection of CEA. The proposed biosensing mechanism for this aptasensor is represented in Figure 1B. We envisioned that the hairpin aptamer structure would unfold upon biorecognition of the analyte, thus unmasking the adamantane residue at the 3′ end of the DNA molecule. This switching mechanism allows supramolecular attachment of the CD-modified HRP enzyme through the formation of host–guest complexes. This hypothesis is based on the high stability of the inclusion complexes of CD with 1-adamantane derivatives, with dissociation constants around 10^−^^4^ M. 

The affinity recognition of the biomarker and further formation of the supramolecular adduct at the electrode surface were studied by cyclic voltammetry. As can be observed in Appendix A, little increase in the redox peak intensities were observed after such incubation processes, suggesting the formation of the affinity and supramolecular complexes on the sensing surface.

### 3.4. Demonstration of the Switching-Based Sensing Mechanism

To further demonstrate the assembly of the host–guest adduct between the adamantane-modified capture aptamer and the HRP-CD conjugate, two additional experiments were performed. First, CEA/Apt-ADA/AuNF/GO/SPE electrodes were incubated with HRP-CD in the absence and the presence of saturated 2-adamantanoic acid sodium salt. The electrodes were then washed and their cyclic voltammetric behavior toward 50 mM H_2_O_2_ in 0.1 M sodium phosphate buffer, pH 7.4, was measured. As is illustrated in Figure 4A, large cathodic currents were measured in the electrode incubated only with HRP-CD, suggesting the redox enzyme remains on the sensing surface. On the contrary, the cyclic voltammogram for the electrode incubated with HRP-CD and the adamantane derivative was similar to those measured for the CEA/Apt-ADA/AuNF/GO/SPE electrode. This fact suggests that the HRP-CD conjugate was not associated to the capture aptamer on the electrode surface due to the prevalent formation of host–guest complexes with the saturating 2-adamantanoic acid salt.

In another set of experiments, the amperometric response of aptasensor after sequential incubation with CEA and either HRP-CD or HRP was determined. The aptasensors not incubated with CEA were employed as control experiments. As is shown in Figure 4B, a noticeable reduction in the cathodic current values was observed for the aptasensor after incubation with CEA and HRP, suggesting that the non-conductive CEA molecules hindered the AuNF surface, thus reducing their electrocatalytic response toward H_2_O_2_, as well as that HRP not being attached to the aptasensor surface. On the contrary, a noticeable amperometric response was observed for the aptasensor after incubation with CEA and HRP-CD, thus demonstrating the switching-based sensing mechanism proposed and the supramolecular attachment of the neoglycoenzyme to the adamantane-modified capture aptamer.

### 3.5. Analytical Performance

The aptasensor was then evaluated for the amperometric determination of CEA, and the optimized working conditions are reported in Appendix A. As is illustrated in Figure 5A, the cathodic amperometric responses increased with the increase in the logarithm of CEA concentrations in buffer solutions, following a linear behavior between 0.1 pg/mL and 1.0 ng/mL CEA. The calibration curve for this determination was adjusted to the following equation (*r^2^* = 0.992, n = 7): *i_c_* (µA) = 0.36·log[CEA] (pg/mL) + 1.41

The limit of detection for this aptasensor, estimated according to the IUPAC rules [21], was calculated as 3.0·10^−^^5^ pg/mL. The aptasensor also showed good reproducibility with a relative standard deviation of 9.3% toward a 20 pg/mL CEA solution (n = 10). This biosensor was highly selective to CEA (Appendix A), and the analytical response was not affected by the presence of potential interfering biomolecules (Appendix A). The aptasensor also retained 85% of the initial sensing capacity after 1 moth of storage at 4 °C in dry conditions (Appendix A).

This biosensor was further tested toward CEA in five-fold diluted reconstituted human serum samples by using the standard addition method, and the resulting calibration curve is shown in Figure 5B. The analytical response showed a linear relationship with the logarithm of CEA concentration between 10 pg/mL and 1.0 ng/mL, according to the following equation (*r*^2^ = 0.979, n = 5):*i_c_* (µA) = 0.37·log[CEA] (pg/mL) + 1.10

Although a short linear range of response was observed in the human serum, no apparent matrix effect was observed in this media, and the limit of detection was estimated as 3.1 pg/mL. This value and those obtained in buffer solutions are lower than the normal cut-off concentrations of CEA in the serum of non-smoking (2.5 ng/mL) and smoking healthy people (5 ng/mL) [22]. The aptasensor was finally validated for the quantification of CEA in three samples of reconstituted human serum spiked with 20 pg/mL of biomarker. The average CEA concentration in the spiked samples was estimated as (19.7 ± 0.4) pg/mL, representing a recovery of 98.5%.

The analytical performance of this aptasensor has been compared with other sensors used for CEA detection (see Appendix A) displaying a very low limit of detection of CEA in human serum samples. Thus, this aptasensor is suitable for CEA detection in clinical samples.

## 4. Conclusions

In this work, we described a novel supramolecular labeling strategy for switching electrochemical aptasensors, based on adamantane-modified aptamers as capture elements and a CD-HRP derivative as a labeling tool. As a proof-of-concept, this approach was successfully evaluated with an electrochemical aptasensor for CEA, constructed on disposable electrodes coated with a highly electrocatalytic GO/AuNF composite nanomaterial. This strategy can be extended to design many other structural switching electrochemical DNA aptasensors for different protein and non-protein analytes, by employing other adamantane-branched specific aptamers and transduction nanomaterials. The easy preparation of CD-based neoglycoenzymes also allows the tailored design of different labeling elements by using other redox enzymes. 

## Data Availability

Not applicable.

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
