# Peer review of "Supramolecular Enzymatic Labeling for Aptamer Switch-Based Electrochemical Biosensor"

_biosensors, 2022, doi:10.3390/bios12070514_

Round 1

Reviewer 1 Report

Villalonga et al reported work " Supramolecular enzymatic labeling for aptamer switch-based electrochemical biosensor" has no significant novelty as compared to reported methods. Additionally, all the techniques used in the current work is conventional without any substantial development in the concept. Therefore I reject this manuscript for the publication in Biosensors  

Author Response

We strongly disagree with this comment. The reviewer do not provided examples of previously published papers dealing with the use of supramolecular interactions allowing enzymatic labelling of aptasensors with switching mechanism.

Reviewer 2 Report

Manuscript ID: biosensors-1747537

Title: Supramolecular enzymatic labeling for aptamer switch-based electrochemical biosensor

In this manuscript a novel labeling strategy for electrochemical aptasensors based on enzymatic marking via supramolecular host-guest interactions was studied by voltammetry and electrochemical impedance. An amperometric aptasensor for carcinoembryonic antigen was assembled and evaluated.

The manuscript is well documented, the experiments are well designed, and the results are very well presented.

The figures and tables support the authors' comments.

Some aspects of the paper need to be improved.

3. Results and Discussion

The section is quite long, and the results presented are difficult to follow. I would recommend splitting into subsections according to the methods used.

Optimization of method parameters (CV, EIS, amperometry) were not presented. A study of the pH or how the amperometry potential was performed is not presented. It is not explained what the connection is between the parameters of the equivalent electrical circuit and the Nyquist diagrams. I suggest you make a comparison with the performance of other sensors used for CEA detection to highlight the advantages of the sensor developed in this paper.

Author Response

REVIEWER 2:

Comments:

In this manuscript a novel labeling strategy for electrochemical aptasensors based on enzymatic marking via supramolecular host-guest interactions was studied by voltammetry and electrochemical impedance. An amperometric aptasensor for carcinoembryonic antigen was assembled and evaluated.

The manuscript is well documented, the experiments are well designed, and the results are very well presented.

The figures and tables support the authors' comments.

Some aspects of the paper need to be improved.

Comment 1: : 3. Results and Discussion

The section is quite long, and the results presented are difficult to follow. I would recommend splitting into subsections according to the methods used.

Response: Thanks to the referee for his/her comments. The manuscript has been revised and subsections have been added according with the guidelines journal.

Comment 2: Optimization of method parameters (CV, EIS, amperometry) were not presented. A study of the pH or how the amperometry potential was performed is not presented .

Response: Thanks to the referee for his/her suggestions. The woring pH and working potential optimization have been included in supporting information in the Table 1S.

Comment 3:  It is not explained what the connection is between the parameters of the equivalent electrical circuit and the Nyquist diagrams.

Response: The connection between the parameters of the Randles equivalent circuit and the Nyquist plot was explained, and this comment was addressed in the manuscript as follow:

Similar results were obtained by EIS, by fitting the experimental data to a conventional Randles equivalent circuit where RS is the solution resistance defining the initial Z´ value, RET is the charge transfer resistance defining the semicircle diameter in the Nyquist plot, CDL is the double layer capacitance and ZW is the diffusional resistance (Warburg element) manifested by the line in the low frequency region.”

Comment 4: I suggest you make a comparison with the performance of other sensors used for CEA detection to highlight the advantages of the sensor developed in this paper.

Response: Thanks to the referee for her/his suggestion. A comparative table of the analytical performance of the aptasensor presented in this work has been included in supporting information as Table 2S. Furthermore, in the manuscript, in the results section we have included the following text:

“The analytical performance of this aptasensor have been compared with other sensors used for CEA detection (See table 2S) displaying very low limit of detection of CEA in Human serum samples. Thus, this aptasensor is suitable for CEA detection in clinical samples.”

Reviewer 3 Report

In this report authors have reported a new switching DNA hairpin based electrochemical aptasensor using a supramolecular labeling strategy. They have used adamantane-aptamer and CD-HRP derivative as a capture element and labeling tool, respectively. Further to test carefully with carcinoembryonic antigen was used, which opens the hairpin and unmask the adamantane reacts with CD. Overall, this is a carefully done study and the findings are of considerable interest and I would recommend publishing this paper. Although there are some issues, which have to clarify before accepting for publish.

- Give full chemical name for HAuCl4 as given for all other chemicals in material and methods section.

- For line 114-117 “the electrode was washed and dried, and 45 μL of 100 mM 114 sodium phosphate buffer, pH 7.4,”  hard to understand. Does the 45 μl of 100 mM sodium phosphate buffer was added to the electrode? and does these solutions made for more than one electrodes as mentioned above or are you talking about different conditions as mentioned in figure 2S? Please clarify it in the manuscript.

-  It will be good if you add a subsection in methods about the amperometric signals- how the signal is measered and how the relative amperometric signal is calculated.

-Typos- supplementary figure 2 and in line 301- change HAS to HSA.

- In figure 2B, what is the control (black bars)? and in line 169- What are 1.2 and 1.5 values, from where they came? are these the ratios of control and CEA, please define in text.

- Reference the actual measurements and figures within the text for ease of understanding, like in the 3rd paragraph at page 6 line 196 “13.8 mm2 after biofunctionalization”- is this the experimental condition as in fig 3 (d)? not only here at other places also.

- With reference to the text “thus unmasking the adamantane residue at the 3 ́ end of the DNA molecule” Although a schematic procedures is shown in text and figure how the Apt-ADA attached to the electrode and the preparation of the adamantane-modified aptamer (Apt-ADA) in methods section. But nowhere earlier discussed at which end of  the hairpin adamentane is attached and then which end of this Apt-ADA complex is attached to the electrode. 

As shown in the material section the 3’-end of hairpin has free -NH2 group, which is attached to the electrode as per scheme 1A, but again with in the same figure it looks like adamentane is attached to the 5’-end as shown by a solid hexagon. 

- In continuation with the previous comment, if I am understanding correctly (which is most likely) the adamentane is placed at the free end of the hairpin. Then how is it masked before biorecognition? as from sequence in material section there is a free chain of six carbons (-CH2 chain) and adamentane can easily be react with CD.

 - CEA is only need to open the hairpin (to show the switching behavior), would it be possible that this switching behavior can also work with response to the ion concentrations? Mg2+ ions can also play an important role as with Mg2+ ions hairpin is more rigid compared to with only Na+ ions. How much ion concentration is used in the experiments, if possible you could check with varying concentrations of ions as control. 

Author Response

REVIEWER 3:

Comments:

In this report authors have reported a new switching DNA hairpin based electrochemical aptasensor using a supramolecular labeling strategy. They have used adamantane-aptamer and CD-HRP derivative as a capture element and labeling tool, respectively. Further to test carefully with carcinoembryonic antigen was used, which opens the hairpin and unmask the adamantane reacts with CD. Overall, this is a carefully done study and the findings are of considerable interest and I would recommend publishing this paper. Although there are some issues, which have to clarify before accepting for publish.

Comment 1: Give full chemical name for HAuCl4 as given for all other chemicals in material and methods section.

Response: Thanks to the referee for his/her suggestion. The tetrachloroauric (III) acid (HAuCl4) has been added in Reagents and instruments section.

Comment 2: For line 114-117 “the electrode was washed and dried, and 45 μL of 100 mM 114 sodium phosphate buffer, pH 7.4,”  hard to understand. Does the 45 μl of 100 mM sodium phosphate buffer was added to the electrode? and does these solutions made for more than one electrodes as mentioned above or are you talking about different conditions as mentioned in figure 2S? Please clarify it in the manuscript.

Response: Thanks to the referee for his/her comments. Once that the electrode surface has been incubated with HRP-CD as last step of the electrochemical sensing strategy, the electrodes are washed and dryed. Then, 45 μL of 100 mM sodium phosphate buffer, pH 7.4 are added as bulk electrolyte solution to the electrochemical interface. This protocol is repeated in the work without making any changes.

It was clarified in the text as follow: “After 1 h incubation at 4ºC, the electrode was washed and dried. To drive analytical measurements, 45 µL of 100 mM sodium phosphate buffer, pH 7.4, were added to the electrochemical cell and amperometric signals were recorded at -200 mV after addition of 5 µL of a freshly prepared 50 mM H2O2 solution.”

Figure 2S refers to the demonstration of the selectivity of this biosensor. Thus, the analytes used were employed during sample incubation step (before HRP-CD incubation), and not after.

Comment 3: It will be good if you add a subsection in methods about the amperometric signals- how the signal is measered and how the relative amperometric signal is calculated.

Response: Thanks to the referee for his/her suggestions. We have added in the experimental part the following text:

“The intensities of the amperometric signals correspond to the different between the stabilized signal and the signal after the addition of H2O2 solution. The relative amperometric signals consider the non-specific signal and it has been calculated as:" (See word document)

Comment 4: Typos- supplementary figure 2 and in line 301- change HAS to HSA.

Response: Thanks to the referee for his/her comments. The typos about Human Albumin Serum (HSA) have been changed.

Comment 5: In figure 2B, what is the control (black bars)? and in line 169- What are 1.2 and 1.5 values, from where they came? are these the ratios of control and CEA, please define in text.

Response:  Thanks to the referee for his/her commets. Control bars in Figure 2B correspond to the assay without the addition of H2O2 which is the substrate of HRP. The 1.2 and 1.5 values show the amperometric response relationship between the use of hydrogen peroxide without hydroquinone against both with bydroquinone (black bars).

Comment 6: Reference the actual measurements and figures within the text for ease of understanding, like in the 3rd paragraph at page 6 line 196 “13.8 mm2 after biofunctionalization”- is this the experimental condition as in fig 3 (d)? not only here at other places also.

Response: Thanks to the referee for his/her suggestions. The measurements and figures have been referenced in the manuscript.

Comment 7: With reference to the text “thus unmasking the adamantane residue at the 3 ́ end of the DNA molecule” Although a schematic procedures is shown in text and figure how the Apt-ADA attached to the electrode and the preparation of the adamantane-modified aptamer (Apt-ADA) in methods section. But nowhere earlier discussed at which end of  the hairpin adamentane is attached and then which end of this Apt-ADA complex is attached to the electrode.

As shown in the material section the 3’-end of hairpin has free -NH2 group, which is attached to the electrode as per scheme 1A, but again with in the same figure it looks like adamentane is attached to the 5’-end as shown by a solid hexagon.

Response: Thanks to the referee for his/her comments. Adamantane is linked to the aptamer through the 3´- because is where the NH2 moeity is. Then, we have edited the scheme 1 according to the mistake represented in the 5’-end of the aptamer: (see word document)

Comment 8: In continuation with the previous comment, if I am understanding correctly (which is most likely) the adamentane is placed at the free end of the hairpin. Then how is it masked before biorecognition? as from sequence in material section there is a free chain of six carbons (-CH2 chain) and adamentane can easily be react with CD.

Response: Thanks to the referee for her/his comments. Naturally, the CEA-aptamer, in absence of CEA, is folded into a hairpin structure that fixed adamantane close to the electrode´s surface. Thus, Adamantane is masked because of the natural conformation of the CEA-aptamer. This process avoids the interaction of HRP-CD without the presence of CEA.

Comment 9: CEA is only need to open the hairpin (to show the switching behavior), would it be possible that this switching behavior can also work with response to the ion concentrations? Mg2+ ions can also play an important role as with Mg2+ ions hairpin is more rigid compared to with only Na+ ions. How much ion concentration is used in the experiments, if possible you could check with varying concentrations of ions as control.

Response: Thanks to the referee for her/his suggestions. The aim of the work is to prepare a selective electrochemical aptasensor against CEA. Thus, we have not used Mg ions to modify the switching behaviour of the hairpin. In addition, it has previously been proved that Mg and Na ions does not affect aptamer conformation at low and physiological concentrations:

  • -J. Tan, et al. Salt Dependence of Nucleic Acid Hairpin Stability. Biophysical Journal 95 (2008) 738–752.
  • Hianik, et al. Influence of ionic strength, pH and aptamer configuration for binding affinity to thrombin. Bioelectrochemistry 70 (2007) 127–133.

Round 2

Reviewer 1 Report

This work does not have any significance and novelty. Just like labelled approach authors demonstrated electrochemical aptasensor. Therefore I reject this manuscript for the publication in Biosensors. 

Reviewer 2 Report

Given the fact that the authors responded to my comments and suggestions, I recommend publishing the manuscript in revised form.

Reviewer 3 Report

The authors have properly addressed all my points and I believe the manuscript is now acceptable.